# The Current Update of Conventional and Innovative Treatment Strategies for Central Nervous System Injury

**DOI:** 10.3390/biomedicines12081894

**Published:** 2024-08-19

**Authors:** Meng-Hsuan Tsai, Chi-Ying Wu, Chao-Hsin Wu, Chun-Yu Chen

**Affiliations:** 1Department of Emergency Medicine, Tungs’ Taichung MetroHarbor Hospital, Taichung 435403, Taiwan; obscurity112358@gmail.com (M.-H.T.); diarrheasoft@gmail.com (C.-Y.W.); t11307@ms3.sltung.com.tw (C.-H.W.); 2Post-Baccalaureate Medicine, National Chung Hsing University, Taichung 40227, Taiwan; 3Department of Nursing, Jen-Teh Junior College of Medicine, Nursing and Management, Miaoli 35664, Taiwan

**Keywords:** traumatic brain injury, spinal cord injury, stem cell therapies, immune modulation

## Abstract

This review explores the complex challenges and advancements in the treatment of traumatic brain injury (TBI) and spinal cord injury (SCI). Traumatic injuries to the central nervous system (CNS) trigger intricate pathophysiological responses, frequently leading to profound and enduring disabilities. This article delves into the dual phases of injury—primary impacts and the subsequent secondary biochemical cascades—that worsen initial damage. Conventional treatments have traditionally prioritized immediate stabilization, surgical interventions, and supportive medical care to manage both the primary and secondary damage associated with central nervous system injuries. We explore current surgical and medical management strategies, emphasizing the crucial role of rehabilitation and the promising potential of stem cell therapies and immune modulation. Advances in stem cell therapy, gene editing, and neuroprosthetics are revolutionizing treatment approaches, providing opportunities not just for recovery but also for the regeneration of impaired neural tissues. This review aims to emphasize emerging therapeutic strategies that hold promise for enhancing outcomes and improving the quality of life for affected individuals worldwide.

## 1. Introduction

Traumatic injuries to the brain and spinal cord represent significant public health challenges affecting millions globally, frequently resulting in severe and enduring impacts on individuals and their families [1]. Neurotrauma predominantly occurs in high-income countries, with males experiencing these injuries approximately 40% more frequently than females [2,3]. According to the Global Burden of Disease Study 2021, an estimated 52.7 million individuals live with the consequences of traumatic brain injury (TBI), while another 42.0 million cope with the effects of spinal cord injury (SCI) [1]. The incidence and burden of these injuries increase significantly after the age of 60 years [1,2,3].

The disparity in morbidity and mortality rates is stark, with nearly 90% of the associated burdens affecting low-income countries [3]. In these regions, the rate of complications from TBI is three times higher than that found in more affluent nations. In the United States alone, TBI is a leading cause of injury-related death, while Europe records over 75,000 TBI-related deaths annually [2,4]. Although TBI accounts for a higher mortality rate, the years lived with disability (YLDs) are notably higher for SCI—5.49 million YLDs for TBI and 4.57 million for SCI globally in 2021 [1,2]. This statistic underscores that, while TBIs are more lethal, SCIs tend to result in more significant long-term disability burdens [1,2].

The prolonged disability associated with SCI underscores the profound impact on quality of life not only for the affected individuals but also for their caregivers and healthcare systems, which often shoulder the responsibility of providing lifelong support and care [5,6]. These statistics not only emphasize the severe impact of CNS injuries but also underscore the critical need for innovative research into new molecular targets and therapeutic strategies. Such advances are crucial not only for improving outcomes but also for alleviating the broader social and economic burdens imposed by these debilitating conditions [1,5,6]. This review aims to explore the paradigm shifts in understanding and treating CNS trauma, with a focus on emerging molecular targets and novel therapeutic strategies that promise to transform the landscape of neurotrauma care.

## 2. Pathogenesis of CNS Trauma

Injuries to the CNS, such as TBI and SCI, initiate a cascade of intricate biological responses [7]. These responses lead to cellular loss, degeneration of axons, and restricted regeneration [8,9,10]. The pathophysiological processes that occur after CNS trauma can be divided into two interrelated phases: primary and secondary injuries [8,9,11]. 

### 2.1. Primary Injury

The primary injury in CNS trauma occurs immediately upon impact, resulting from external mechanical forces such as direct impacts, sudden changes in velocity, penetrating injuries, or the effects of explosive blast waves [8]. These forces cause direct and immediate structural damage to the tissues of the brain and spinal cord, profoundly disrupting their functional integrity [7,8,9]. Key manifestations of primary injury include the following.

#### 2.1.1. Focal Contusions and Hematomas

These are localized areas of bleeding and bruising within the CNS that directly damage brain or spinal cord tissues. Such injuries can lead to compression of surrounding neural structures and substantial disruption of local blood flow, further exacerbating the damage [8,9].

#### 2.1.2. Diffuse Axonal Injury (DAI) 

Resulting from shearing forces during traumatic events, DAI entails extensive disruption of white matter tracts in the brain. This disruption results in a breakdown of neuronal connections across diverse brain regions, profoundly impairing neural communication and overall brain function. The consequences of DAI can vary from subtle cognitive impairments to severe neurological disabilities [12,13,14]. 

#### 2.1.3. Swelling 

Trauma-induced swelling can manifest both as focal, affecting specific areas, and diffuse, impacting broader regions of the CNS. This swelling raises intracranial pressure, which can exacerbate damage to delicate neural tissues and disrupt the flow of essential nutrients and oxygen crucial for brain function. Often, this swelling aggravates the initial trauma, resulting in more extensive and prolonged functional impairments [8,10,13,15].

Following these immediate structural damages, a robust inflammatory response is initiated [9]. This response is characterized by the activation of microglia, which serve as the primary immune defense cells in the CNS [9,16]. Additionally, there is substantial infiltration of peripheral immune cells into the CNS and the release of various inflammatory mediators [9]. While these responses are initially protective, they can also become detrimental, leading to further tissue loss, the formation of scar tissue, and significant impediments to the natural regenerative processes of the nervous system [8,17]. This intricate interplay between damage and defense mechanisms complicates the path to recovery and regeneration following CNS trauma [8].

### 2.2. Secondary Injury 

Secondary injury unfolds in the hours to months following the initial trauma, involving a complex cascade of biochemical and redox reactions that worsen the primary damage [7,8]. This phase is characterized by the following.

#### 2.2.1. Inflammatory Response 

Shortly after the primary injury, resident immune cells in the CNS, particularly microglia, undergo activation. Transitioning from a quiescent to an activated state, they release cytokines and chemokines that trigger additional immune responses. This recruitment includes peripheral immune cells like neutrophils and macrophages to the site of injury. While essential for debris clearance, these cells can also release harmful substances that damage surrounding tissues [9,18,19]. Cytokines such as TNF-α, IL-1β, and IL-6, along with chemokines like CCL2, play critical roles in regulating the inflammatory response. While these molecules are pivotal in initiating healing processes, their prolonged or excessive presence can induce sustained inflammation, resulting in additional cell death and tissue damage [11,20,21].

#### 2.2.2. Oxidative Stress 

Secondary injury is notably characterized by the excessive production of reactive oxygen species (ROS), which are chemically reactive molecules including free radicals. Normally generated in small quantities during mitochondrial respiration, their levels surge significantly following trauma, causing damage to cellular structures such as membranes, DNA, and proteins, ultimately resulting in cell death [11,21]. The surge in ROS often overwhelms the body’s natural antioxidant defenses, resulting in oxidative stress. This imbalance significantly contributes to structural damage at the cellular and subcellular levels in neurons and other cell types within the CNS [21].

#### 2.2.3. Excitotoxicity 

TBI leads to the excessive release of glutamate, the brain’s primary excitatory neurotransmitter. Normally, glutamate facilitates neurotransmission, plasticity, and memory formation. However, excessive release can overactivate glutamate receptors such as NMDA and AMPA receptors, causing an influx of calcium ions into neurons [22,23,24]. Elevated intracellular calcium levels activate various enzymatic processes that can be detrimental to cell health, including phospholipases, endonucleases, and proteases. Additionally, mitochondrial dysfunction is triggered, collectively contributing to neuronal death [21,24].

#### 2.2.4. Apoptosis and Necrosis 

Following CNS injury, cells may undergo apoptosis (programmed cell death) or necrosis (cell death caused by injury or disease). The balance between these two types of cell death can profoundly influence the outcome of the injury. Apoptosis is typically viewed as a more controlled form of cell death that minimizes inflammation, whereas necrosis can induce inflammation and exacerbate damage to surrounding tissue [24,25,26,27].

#### 2.2.5. Blood–Brain Barrier (BBB) Disruption 

After trauma, the integrity of the BBB is compromised, permitting cells and proteins that are typically restricted from the brain to enter the neural environment. This can lead to increased edema, inflammation, and potentially the introduction of pathogens, which may result in infection [25].

The secondary injury phase often leads to progressive degeneration, with tissue loss frequently exceeding that observed in the primary injury phase [8,18]. Table 1 provides a comparison of primary and secondary injuries. This highlights a critical therapeutic window where timely and targeted interventions may potentially halt or even reverse the progression of damage [8]. Understanding these interconnected phases of primary and secondary injuries is crucial for developing effective therapeutic strategies. Early surgical interventions are crucial for managing lesions from primary injuries and stabilizing patients [28,29]. Concurrently, ongoing detection, prevention, and management of secondary injuries are pivotal in neuro-intensive care, aiming to minimize long-term disabilities and enhance functional recovery [24].

### 2.3. The Classification and Severity of CNS Trauma

#### 2.3.1. TBI Are Classified Into Three Categories Based on Severity

Mild TBI is characterized by a brief or no loss of consciousness, post-traumatic amnesia for less than 1 h, and an initial Glasgow Coma Scale (GCS) score of 13 to 15. Patients might experience headaches, confusion, dizziness, and may have normal imaging results.

Moderate TBI involves loss of consciousness for more than 30 min but less than 24 h, post-traumatic amnesia lasting 1 to 24 h, and an initial GCS score of 9 to 12. Symptoms are more persistent, and CT scans or MRIs may show visible brain damage.

Severe TBI is defined by a loss of consciousness for more than 24 h, post-traumatic amnesia for more than 24 h, and an initial GCS score of 3 to 8. Patients often exhibit significant cognitive and neurological deficits, with imaging revealing extensive brain damage.

The severity of TBI guides initial treatment and influences prognosis, with long-term outcomes varying significantly among individuals [4,7,14].

#### 2.3.2. SCI Common Standardized and Classified System 

The American Spinal Injury Association (ASIA) Impairment Scale is a standardized system used to classify the severity of spinal cord injuries (SCI). It is based on sensory and motor function assessments following the injury [30]. Here is how each grade on the ASIA scale describes the severity of an SCI:ASIA A (Complete): No motor or sensory function is preserved in the sacral segments S4–S5. It indicates a complete spinal cord injury where no motor or sensory function is preserved below the level of injury, including the anal and perineal area.ASIA B (Incomplete): Sensory but not motor function is preserved below the neurological level of injury and extends through the sacral segments. This grade reflects some preservation of sensory function below the injury level, with no motor function below the level.ASIA C (Incomplete): Motor function is preserved below the neurological level, and more than half of key muscle functions below the neurological level have a muscle grade less than 3. It indicates partial preservation of motor functions, where some movements are possible but not strong enough to overcome gravity.ASIA D (Incomplete): Motor function is preserved below the neurological level, and at least half of key muscle functions below the neurological level have a muscle grade of 3 or more. It represents a less severe form of incomplete injury, with more significant preservation of motor function and ability to move against gravity.ASIA E (Normal): Motor and sensory function are normal. The individual has regained normal function, though they may have had a spinal cord injury previously or had transient deficits.

The ASIA scale helps predict potential recovery outcomes. Patients with a higher grade on the scale (e.g., ASIA D) at initial assessment often have a better prognosis for functional recovery than those with lower grades (e.g., ASIA A or B) [30].

## 3. Treatment Strategies of CNS Injury

The management of TBI and SCI poses complex challenges necessitating a comprehensive therapeutic approach (Figure 1). Historically, conventional treatments have focused on immediate stabilization, surgical interventions, and supportive medical care to address both the initial and secondary damage associated with CNS injuries [29,31]. These traditional strategies are fundamental to patient recovery and encompass decompressive surgeries, intracranial pressure monitoring, and extensive rehabilitation programs aimed at restoring function and improving quality of life [32]. In recent years, the emergence of innovative therapies has brought new hope and expanded treatment options [33]. Breakthroughs in stem cell therapy, gene editing, and neuroprosthetics are transforming treatment strategies, offering prospects not only for recovery but also for the regeneration of damaged neural tissues [16,33,34]. Integrating these novel approaches with traditional methods holds promise for enhancing outcomes and reshaping recovery pathways for patients with these challenging conditions [35]. Therefore, an interdisciplinary approach is crucial for the effective management of CNS trauma, combining state-of-the-art technology with established medical practices to achieve optimal results [35].

### 3.1. Current Approaches

#### 3.1.1. Surgical Treatment

Decompressive craniectomy is indicated for alleviating increased intracranial pressure (ICP) that does not respond to medical treatment. This procedure is recommended in cases of severe brain swelling or large mass lesions causing substantial brain displacement. It involves removing a portion of the skull to facilitate brain expansion, thereby reducing ICP and enhancing blood flow [28,29,36]. Emergency craniotomy may be performed to evacuate hematomas such as subdural (SDH) or epidural hematomas (EDH). Surgical intervention is typically recommended when there is significant mass effect from the hematoma, characterized by neurological deterioration or radiological evidence indicating severe compression of brain structures [28,29,36,37,38,39,40,41]. Invasive monitoring of ICP is essential for managing patients with severe TBI, guiding both medical and surgical interventions. ICP monitoring plays a critical role in determining the necessity for further surgical procedures and optimizing strategies to manage cerebral perfusion pressure [28,37,42,43]. External ventricular drains (EVD) are utilized for diagnostic and therapeutic purposes, facilitating cerebrospinal fluid (CSF) drainage to control ICP and allowing for CSF sampling for analysis [38,39,44,45]. In patients with penetrating trauma, surgical management often includes debridement, removal of foreign objects, and repair of dural and skull defects. This approach is especially crucial in cases of gunshot wounds or other penetrating injuries, where the goals include managing brain herniation and minimizing secondary brain injury. Additionally, prophylactic antibiotic administration is typically recommended to prevent infections [46,47,48].

#### 3.1.2. Medical Treatment

Pharmacological treatment is pivotal in the management of TBI, with a focus on reducing intracranial pressure, mitigating secondary damage, and enhancing neuroprotection [34,49]. Optimal medication strategies encompass the administration of osmotic diuretics, antiseizure medications, and neuroprotective agents, aiming to stabilize and support the brain’s recovery process [49,50,51]. These interventions are essential not only for immediate stabilization but also for preventing long-term complications and enhancing overall outcomes, underscoring their pivotal role in comprehensive TBI management [34,49,50,51].

Mannitol and hypertonic saline are primary osmotic agents utilized to alleviate cerebral edema and decrease ICP. Additional methods such as sedation, paralysis, and CSF drainage are employed to actively manage ICP [51,52,53]. Effective management of ICP is critical for preventing secondary ischemic injuries and enhancing overall outcomes, as elevated ICP can exacerbate brain damage if not properly controlled [52,53]. Neuroprotection involves administering medications that stabilize neuronal cell membranes, reduce metabolic demand, and mitigate oxidative stress. These strategies are crucial in minimizing neuronal damage following TBI. Careful management of body temperature and blood glucose levels is essential to prevent worsening brain injury. Additionally, antioxidants and other agents are utilized to protect against ongoing cellular damage [51,54,55]. Post-traumatic seizures are a frequent complication of TBI and can exacerbate brain damage by raising metabolic demands. Preventing these seizures is crucial for safeguarding the brain during its recovery phase [51,56,57,58]. Antiepileptic drugs such as phenytoin and levetiracetam are frequently prescribed to reduce the risk of seizures in the acute phase following a brain injury [51,56,57,58]. To ensure adequate blood flow to the brain, which is essential for delivering necessary oxygen and nutrients to injured neuronal tissues. This may involve the use of vasopressors to maintain systemic blood pressure at a level that supports cerebral perfusion. Fluid management is also crucial for maintaining blood volume and pressure, thereby facilitating proper cerebral blood flow [51,59,60,61]. Managing body temperature is crucial, as hyperthermia can increase the risk of additional brain injury. Cooling measures may be necessary to maintain normothermia [51,54,55,56,57,58,59,60,61,62]. Additionally, ensuring adequate nutrition is crucial for supporting the heightened metabolic demands of the injured brain. This often entails the use of specialized nutritional formulas administered via enteral or parenteral feeding to ensure nutritional requirements are met [54,63]. Research into new drugs aimed at modulating the immune response, reducing inflammation, and potentially promoting neuronal regeneration is actively ongoing. This includes trials with monoclonal antibodies designed to target specific harmful pathways activated following TBI [51,64]. Effective management entails not only addressing the primary damage caused by the injury but also mitigating the secondary processes that can prolong recovery and worsen outcomes. This integrated treatment strategy ensures a holistic approach to care, which is essential for enhancing recovery and improving the quality of life for individuals affected by TBI.

#### 3.1.3. Rehabilitation

Rehabilitation incorporates a multidisciplinary team of healthcare professionals, including nurses, physicians, dieticians, psychologists, physiotherapists, social workers, recreation therapists, speech therapists, orthotists, and child life specialists. This collaborative approach ensures comprehensive care that addresses all aspects of a patient’s health, encompassing both physical and psychological needs [35,65]. Physical rehabilitation focuses on restoring lost functions, enhancing remaining functions, and preventing secondary complications [35,66,67]. Key components include strength training, cardiovascular exercises, respiratory conditioning, mobility training, and stretching exercises [67,68]. These activities not only enhance physical capabilities but also have been demonstrated to induce significant changes in cellular signaling and growth factor expression, potentially facilitating further recovery [69,70].

#### 3.1.4. Weight-Supported Locomotor Training (WSLT)

WSLT utilizes devices to assist the patient’s weight during locomotion exercises on treadmills or open ground. This approach aims to restore neural pathways affected by spinal injuries and enhance mobility and cardiorespiratory health. While studies have demonstrated its effectiveness, challenges persist in its broader adoption due to resource constraints [35,71].

#### 3.1.5. Occupational Therapy

Occupational therapy in rehabilitation emphasizes assisting patients in integrating adaptive devices into their daily routines to optimize functional independence. This includes incorporating wheelchairs, lifts, specialized bathroom equipment, and vehicle modifications, all essential for reintegrating into community life and sustaining independence [35,72].

#### 3.1.6. Functional Electrical Stimulation (FES)

FES is utilized to stimulate muscles with electrical pulses, aiding in the restoration of movement in affected limbs. It has proven effective in tasks such as eating, gripping objects, and ambulation, facilitated by devices like the Parastep for walking assistance and stationary bicycles for lower limb exercise [73,74].

### 3.2. Innovative Therapies

#### 3.2.1. Stem Cell Therapies

Stem cell therapy for the CNS represents a pioneering frontier in regenerative medicine, addressing a spectrum of debilitating conditions such as SCI, Parkinson’s disease, multiple sclerosis, and Alzheimer’s disease [11,33]. The CNS’s limited capacity for self-repair significantly influences the specificity and efficacy of treatments like stem cell therapy, highlighting the crucial need for precise and targeted therapeutic strategies to promote regeneration and functional recovery in CNS injuries [33,65,75]. Stem cells, capable of differentiating into various types of neural cells, offer a promising approach to overcoming the inherent regenerative challenges of the CNS, thereby serving as a pivotal tool in advancing treatments for these complex injuries and potentially transforming neurological rehabilitation and care [75,76].

In CNS disorders, damage typically results in the loss of neurons and glial cells, along with disruptions in neural circuits critical for functionality [77]. Stem cell therapy aims to address these issues by introducing cells capable of differentiating into both neurons and glial cells, thereby replacing lost or damaged tissue and promoting regenerative processes [33,75,77]. This therapy holds transformative potential for treating neurodegenerative diseases through various therapeutic mechanisms. One primary strategy involves cell replacement, where stem cells are implanted into affected brain regions to differentiate into specific types of neurons [78,79]. In addition to replacing lost cells, stem cells contribute to a supportive environment conducive to brain repair [80,81]. They secrete growth factors such as brain-derived neurotrophic factor (BDNF) and nerve growth factor (NGF), which promote neuron survival and growth while inhibiting apoptosis [81,82,83]. This dual action not only aids in neural repair but also modulates inflammatory responses, which is crucial in managing conditions like Alzheimer’s disease, where inflammation significantly influences disease progression [84]. 

The strategies employed in CNS stem cell therapy are diverse. They involve transplanting neural stem cells that mature into neurons and other supportive cells and utilizing mesenchymal stem cells known for their neuroprotective effects through the secretion of trophic factors [85,86]. These factors contribute to reducing inflammation and promoting tissue survival [87,88]. Induced pluripotent stem cells have also gained prominence, as they can be derived from a patient’s own cells, thereby reducing the risks of immune rejection and addressing ethical concerns associated with embryonic stem cells [79,89,90,91]. Neural stem cells (NSCs) are valued for their ability to self-renew and differentiate into various CNS cell types, including neurons and glial cells [33,77,80,92]. This capability has been harnessed to enhance functional recovery following SCI by promoting neurogenesis and plasticity while reducing neuroinflammation [87]. However, the clinical application of NSCs encounters challenges such as risk of tumor formation if the NSCs proliferate uncontrollably and low survival rates attributed to ischemia and immune rejection [77,88]. In response, researchers are exploring small extracellular vesicles derived from NSCs (NSC-sEVs) for their potential in SCI regeneration, showing promising outcomes in reducing neuroinflammation, mitigating neuronal apoptosis, and promoting autophagy. But we still had limited data on long-term survival and functionality of NSCs post-transplantation [88].

Mesenchymal stem cells (MSCs), derived from bone marrow, adipose tissue, or umbilical cord blood, play a crucial role in SCI therapy due to their capability to differentiate into various cell types, anti-inflammatory properties, and immune-modulating abilities. They have shown potential in improving motor and sensory functions in SCI patients in clinical trials. [85,93,94,95,96,97,98]. A significant advantage of MSCs is their capacity to cross the BBB, enabling them to deliver therapeutic effects directly to affected areas of the brain without invasive procedures [85,86,99]. Ongoing clinical trials are evaluating the safety and efficacy of MSCs in SCI treatments, with some patients demonstrating notable improvements in function and quality of life. However, the effectiveness can be transient, requiring repeated administrations for sustained benefits. The potential side effect of MSCs are risk of infection from the transplantation procedure and potential for ectopic tissue formation if MSCs differentiate into unintended cell types [75,100]. Human induced pluripotent stem cells (hiPSCs) can be reprogrammed from adult cells and possess the capacity to differentiate into any cell type in the body [78,79,89]. These cells have been employed to generate neural stem/progenitor cells for transplantation into injured spinal cords, demonstrating promising outcomes in initial studies focused on enhancing motor functions and reducing lesion sizes following SCI. But there is a risk of tumorigenesis due to the pluripotency of these cells. [79,100,101]. Embryonic stem cells (ESCs), renowned for their pluripotent capabilities, can differentiate into any cell type, including oligodendrocytes crucial for the remyelination of neurons [91,102,103]. Despite ethical considerations, their effectiveness in promoting regeneration and functional recovery in SCI models has been validated through numerous studies. We have to watch out for the high risk of immune rejection unless matched very carefully. [91,103]. Exosomes, particularly those derived from MSCs and NSCs, are under investigation for their regenerative potential [98,104,105]. These exosomes contain a variety of biomolecules, including nucleic acids and proteins, which promote tissue repair and recovery following CNS injuries. Exosomes derived from stem cells are collected and purified to be used as a vehicle for delivering growth factors, mRNA, and miRNA that can aid in repairing injured spinal cord tissue [22,98,104]. They also confer anti-inflammatory and neuroprotective effects, potentially facilitating functional recovery and serving as biomarkers for SCI [98].

Currently, various types of stem cells are being investigated for their application in TBI (Table 2) [86,98,101,102,103,104,105,106,107]. Stem cell therapy represents a significant and evolving area of research with the potential to substantially improve outcomes for individuals with CNS injuries. While advancements are being made, further research is necessary to fully comprehend the capabilities and limitations of these therapies. Innovative approaches, including encapsulating stem cells in biomimetic hydrogels and developing 3D-printed scaffolds and biomimetic materials, are being explored to enhance the therapeutic effects of stem cell transplantation [75]. Despite its promise, stem cell therapy for CNS disorders remains largely experimental, facing ongoing challenges such as ensuring the safety and stability of transplanted cells, controlling their differentiation in vivo, and achieving functional integration into existing neural networks [9,19,25].

#### 3.2.2. Modulation of Inflammatory and Immune Responses

The CNS has a distinctive immune environment known as “immune privilege”, which means it is somewhat isolated from the body’s broader immune system. However, trauma can compromise this immune privilege, triggering a cascade of inflammatory responses that may exacerbate the injury. Microglia are the resident macrophages of the CNS and play a pivotal role in the initial immune response to injury. In their quiescent state, microglia perform maintenance and support functions [108]. However, following injury, they can become activated and release proinflammatory cytokines that contribute to neuronal damage. Modulating microglial activation involves shifting their phenotype from a proinflammatory state to a reparative, anti-inflammatory state. This can be achieved through the administration of specific cytokines, pharmacological agents, or even certain neuropeptides that promote microglial support for repair, such as enhancing debris clearance and the release of growth factors [76]. Cytokines such as tumor necrosis factor-alpha (TNF-α), interleukins (such as IL-1β and IL-6), and others play crucial roles in the inflammatory response following CNS trauma. These molecules can exacerbate injury through various pathways, including promoting immune cell infiltration to the injury site, which can further damage neural tissues [12,21,87]. Utilizing cytokine inhibitors or neutralizing antibodies to block the activity of specific cytokines has demonstrated efficacy in reducing inflammation and safeguarding neural tissue. These treatments are instrumental in limiting the secondary damage that occurs after the initial injury phase [87,109,110]. Chemokines constitute another class of signaling proteins that attract cells, especially immune cells, to sites of inflammation. In CNS trauma, chemokines can result in excessive infiltration of immune cells, leading to harmful consequences [25,110]. Antagonists or inhibitors of chemokine receptors can be utilized to prevent this excessive migration of immune cells to the CNS. This approach aids in mitigating the intensity of the inflammatory response and in preserving neural function [11,110].

#### 3.2.3. Combination Therapies

Combining different therapeutic approaches can synergistically enhance recovery. This could entail integrating physical therapies with biological treatments. For instance, integrating physical rehabilitation with stem cell therapy could enhance both the physical retraining of motor functions and biological recovery at the neuronal level. Additionally, using pharmacological agents to prepare the injured CNS environment for the integration of transplanted stem cells or to support endogenous repair processes may further improve outcomes [85,100].

## 4. Clinical Implications and Future Directions in CNS Injuries

Effectively modulating the inflammatory and immune responses in CNS trauma can profoundly influence patient outcomes by mitigating secondary damage, fostering recovery, and potentially improving long-term functional results. Future research aims to elucidate the intricate balance of the immune system within the CNS, develop targeted therapies capable of precisely modulating this system without compromising its crucial protective functions, and integrate these strategies into comprehensive treatment protocols that address both biological and functional aspects of recovery. These additional explanations offer a more comprehensive understanding of how inflammation and immune responses can be effectively modulated in CNS trauma. They underscore the intricate interplay of biological processes that can be leveraged to enhance therapeutic outcomes.

## 5. Limitations in the Treatment of CNS Injuries

Despite the significant advancements in CNS trauma treatment discussed, this review acknowledges several limitations. Variability in treatment outcomes across different demographics and socioeconomic groups may impact the generalizability and effectiveness of interventions. The translation of novel therapies from clinical trials to standard care is often hindered by regulatory, logistical, and ethical challenges. Furthermore, the predominance of data from high-income countries may not fully reflect global realities, potentially limiting the applicability of findings worldwide. Moreover, the complexity of CNS injuries necessitates increasingly personalized treatment approaches, a goal that remains partially unrealized due to existing technological and scientific constraints.

## 6. Discussion on Challenges in Clinical Translation

While this review effectively outlines the potential of novel therapies, a critical gap remains unaddressed—the challenges in translating these experimental therapies into clinical practice. This gap is crucial, as the translation of experimental medicine to routine clinical application involves significant hurdles that are often underexplored in academic discussions.

### Bridging Experimental and Clinical Realms

Regulatory Challenges: Many promising therapies encounter regulatory barriers that delay approval processes, limiting rapid integration into clinical use. Regulatory bodies demand extensive data on safety and efficacy, which requires prolonged and costly clinical trials.Scalability and Accessibility: Even if a therapy shows promise in controlled experimental settings, scaling it to be widely accessible poses logistical and manufacturing challenges. The cost implications are often prohibitive, affecting the equitable distribution of these advanced treatments.Ethical Concerns: Particularly with therapies like stem cell treatment, ethical debates can impede progress. Issues around stem cell sourcing and the long-term impact of genetically modified cells in humans need resolution within ethical frameworks before such treatments can become mainstream.Long-Term Efficacy and Safety: Many experimental treatments lack long-term data, making it difficult to predict their future impact. Clinicians and patients may be wary of potential long-term complications or side effects that have not yet been fully understood.

In conclusion, while this review outlines significant advancements in the treatment of CNS injuries, addressing the challenges of clinical translation is imperative for these innovations to truly impact patient care. By acknowledging and strategizing around these challenges, the gap between experimental success and clinical application can be narrowed, potentially transforming the prognosis for individuals with TBI and SCI.

## 7. Conclusions

The management of CNS injuries, including both TBI and SCI, has undergone significant evolution with advancements in medical technology and a deeper understanding of injury pathophysiology. Surgical and medical treatments have become more refined, and the role of rehabilitation in recovery is increasingly recognized, underscoring the importance of a multidisciplinary approach. Moreover, emerging therapies such as stem cell treatments and immunomodulation offer promise for further enhancing patient care. However, future research should prioritize improving the precision of these interventions, ensuring their accessibility, and integrating them into comprehensive treatment paradigms. These efforts are essential for shifting the current treatment landscape towards more effective, personalized, and regenerative solutions, ultimately reducing the global burden of these debilitating conditions.

## Figures and Tables

**Figure 1 biomedicines-12-01894-f001:**
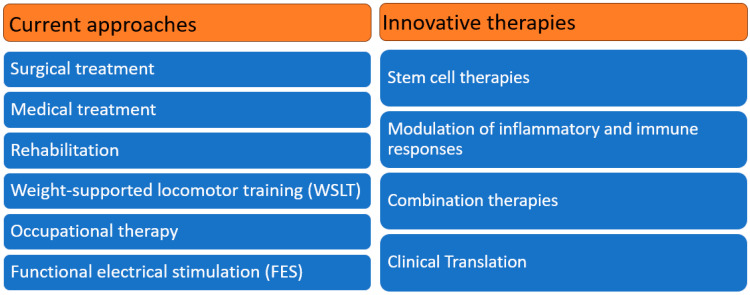
Treatment strategies for current approaches and innovative therapies in CNS injuries.

**Table 1 biomedicines-12-01894-t001:** Summary of the current comprehension of primary and secondary injuries in the context of TBI and SCI.

	Primary Injury	Secondary Injury
Timing	Immediate	Delayed (minutes to months after trauma)
Causes	External mechanical forces (impact, penetration, etc.)	Biochemical and cellular response to primary injury
Pathophysiology	Physical disruption of CNS tissue	Inflammation, oxidative stress, excitotoxicity, BBB disruption
Clinical Manifestations	Contusions, hematomas, DAI, tissue swelling	Cerebral edema, ischemia, cell death
Treatment Objectives	Physical disruption of CNS tissue	Physical disruption of CNS tissue

CNS: central nervous system, BBB: blood-brain barrier, DAI: Diffuse Axonal Injury.

**Table 2 biomedicines-12-01894-t002:** The comparative classification of various stem cells.

Stem Cell Type	Source	Key Characteristics	Therapeutic Actions	Clinical Application
Mesenchymal Stem Cells (MSCs)	Bone marrow, adipose tissue, dental tissues, etc.	Low immunogenicity, multipotent, can differentiate into various tissue types	Can differentiate into cell types that replace damaged tissue, secrete anti-inflammatory and regenerative factors	Numerous clinical trials, showing potential to improve functional outcomes, safe and potentially effective
Exosomes and Extracellular Vesicles	Derived from MSCs and other cell types	Contain proteins, lipids, and nucleic acids that facilitate intercellular communication	Transfer regenerative molecules to injured cells, promote neurogenesis and angiogenesis	Research focused on their role in modulating inflammation and delivering growth factors for tissue repair and regeneration
Neural Stem Cells (NSCs)	Derived from embryonic or adult neural tissues	Can differentiate into neurons, astrocytes, and oligodendrocytes	Integrate into the host spinal cord, replace lost neurons and glial cells, reduce inflammation, support recovery	Preclinical and some early clinical studies show promise in enhancing recovery of motor functions
Human Induced Pluripotent Stem Cells (hiPSCs)	Adult cells reprogrammed to an embryonic stem-cell-like state	Pluripotent, can differentiate into any cell type, personalized therapy option with reduced rejection risks	Differentiate into neural cells to promote nerve regeneration, reduce lesion size	Experimental stages; promising results in animal models for improving outcomes in SCI
Embryonic Stem Cells (ESCs)	Derived from preimplantation embryos	Pluripotent, capable of differentiating into any cell type	Differentiate into oligodendrocytes for remyelination of neurons, essential for SCI repair	Ethical concerns limit use, but research continues under controlled settings for potential in treating severe SCI

## Data Availability

No new data were created or analyzed in this study. Data sharing is not applicable to this article.

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
