# Peer review of "The Current Update of Conventional and Innovative Treatment Strategies for Central Nervous System Injury"

_biomedicines, 2024, doi:10.3390/biomedicines12081894_

Round 1

Reviewer 1 Report

Comments and Suggestions for Authors

The reviewer would like to declare no conflict of interest with the authors and their associated institutions as stated in the title page.

1. Table 2, Is the information in the table being referred to from another piece of literature? If yes, kindly cite the relevant papers.

2. The author should be careful on the use of abbreviation. Once the abbreviated was used to refer to a long name (eg CNS for central nervous system), the subsequent mention of the term should use the abbreviation. Many instances were sighted, where the abbreviations were defined in an earlier section, but still the full term was being used. Then what's the point of setting an abbreviation?

3.  Section 3.1, this section (and subsections) sumarised the current therapies available for CNS injury. However, since these disorders (TBI and SCI) are not uncommon, the authors should cite or take reference to the current best practices or guidelines on the management of TBI and SCI. 

4. The term "development" in the title suggests that the article would be describing a chronological sequence of events for the treatment of CNS injury, however this is not reflected in the manuscript. Suggest to be replaced with the term "current update".

5. The term "traditional" in the title suggests traditional ethnological treatment, which the article did not cover this part.. suggest this term is to be replaced with "conventional".

6. The term "Contemporary" in the title should be replaced with "Innovative" 

Author Response

As Word file.

Reviewer 2 Report

Comments and Suggestions for Authors

Tsai et al. investigate the intricate obstacles and developments in the management of spinal cord injury (SCI) and traumatic brain injury (TBI). The paper explores the two phases of damage, primary effects and later secondary biochemical cascades, that aggravate first harm. It addresses conventional therapies, modern surgical and medical management techniques, and stresses the critical function of rehabilitation. Aiming to improve results and thus the quality of life for affected people worldwide, the study also emphasizes the prospective possibilities of stem cell therapies, immunological modulation, gene editing, and neuroprosthetics in transforming treatment approaches.

Nevertheless, this work necessitates substantial revisions to resolve a number of critical issues:

1. Although traditional treatments are discussed, the review does not go into great depth on present best practices in TBI and SCI management—that is, those described in clinical guidelines.

2. Particularly the new medications, the study does not critically assess the efficacy or constraints of the stated remedies. A thorough review depends on a balanced examination.

3. The paper notes "intricate pathophysiological responses" without going into great depth on these mechanisms. Understanding therapeutic strategies depends on thorough knowledge of pathophysiological processes.

4. Important elements of TBI and SCI management—such as the use of standardized evaluation techniques (e.g., Glasgow Coma Scale for TBI) or damage classification systems—that the review does not address.

5. Although rehabilitation is underlined as vital, the study does not go into much detail on certain rehabilitation techniques or their relevance in long-term healing.

6. Two currently uninformative tables in the review need to be strengthened with more thorough data; they represent lack of information on pathogenesis. Furthermore lacking are figures showing the course of TBI and SCI pathogenesis. Combing the tables with thorough information on treatment results, modes of action, and relative efficacy would improve their value. The clarity and instructional worth of the review would be much enhanced by include figures that graphically show the main and secondary damage mechanisms as well as the biochemical cascades engaged.

7. Lack of Discussion on Challenges in Clinical Translation: A major concern in the field, the review does not address the problems in transferring exciting experimental medicines to clinical practice.

Tsai et al. should give a more thorough explanation of existing best practices in management, critically assess the potential of new treatments, and include visual aids to better understanding to help to improve the review. Furthermore enhancing the comprehensiveness and applicability of the study to clinical practice would be addressing the difficulties in clinical translation of experimental medicines and offering more thorough knowledge on rehabilitation procedures.

Comments on the Quality of English Language

Minor editing of English language required

Author Response

As word file.

Round 2

Reviewer 1 Report

Comments and Suggestions for Authors

The authors have addressed my previous comments. I have no further comments.

Reviewer 2 Report

Comments and Suggestions for Authors

The revision has significantly improved the manuscript, which is now suitable for publication.